# UnAC: Evoking Complicated Multimodal Reasoning in LMMs

## Abstract

The recent large multimodal models (LMMs) have demonstrated their impressive capability of image understanding. However, they still struggle to make complicated reasoning for solving a challenging multimodal problem. In this paper, we present UnAC (Understanding, Abstracting, and Checking), a novel multimodal prompting method, to synergize reasoning for complicated problems in the multimodal context of LMMs, such as GPT-4o, Gemini-1.5 and GPT-4V. To improve the understanding of the image and capture more details, we propose an adaptive visual prompting method to make LMMs able to focus on certain regions. An image abstracting prompting is designed to effectively extract information from images. Further, we propose a gradual self-checking scheme for leading to better reasoning by checking each decomposed sub-question and its answer. Extensive experiments on three public benchmarks – MathVista, MM-Vet, and MMMU – demonstrate the effectiveness of our method.

## 1 Introduction

In recent years, large language models (LLMs) have advanced significantly Brown et al. (2020); Achiam et al. (2023); Touvron et al. (2023); Bubeck et al. (2023); Chowdhery et al. (2023); Zhang et al. (2022). From GPT-3 Brown et al. (2020), PaLM Chowdhery et al. (2023) and Llama Touvron et al. (2023) to GPT-4 Achiam et al. (2023) and PaLM-2 Anil et al. (2023). Notably, Generative Pre-trained Transformers (GPTs) Brown et al. (2020); Achiam et al. (2023) have driven numerous breakthroughs in both industry and academia. Since the release of GPT-4, there has been increasing interest in large multimodal models (LMMs) within the research community. Many approaches are focused on developing powerful multimodal models based on open-source frameworks Liu et al. (2024); Wu et al. (2023); Dai et al. (2024); Zhu et al. (2023). Recently, the release of GPT-4V(ision) and Gemini-1.5-flash Team et al. (2023) has garnered immediate attention for its impressive capability of understanding images. However, they still struggle to do some complicated multimodal reasoning tasks Lu et al. (2023); Yue et al. (2023).

Since approaches Yao et al. (2024); Wei et al. (2022); Yao et al. (2022); Miao et al. (2023); Zheng et al. (2023) of prompting to improve the reasoning ability with LLMs in only language-context make significant progress. However, since LMMs can not able to decompose an image easily like decomposing a sentence, it is ineffective to apply the language prompts to improve reasoning in the visual context. For answering a question in the visual context, the major failure cases are due to the misunderstanding of the image or imprecisely summarizing the information. The reason for missing or misunderstanding some details is related to the weak capability of getting fine-grained information Yang et al. (2023a). Visual prompts have also been explored for various multi-modal tasks, especially for enhancing the performance of fine-grained visual tasks. Those methods focus on encoding some masks like points, boxes, and lines combined with the input features or directly applying overlays on the original image. Most recently, Yang *et al.* proposed to build the visual prompting mechanism by partitioning the image into a set of semantically meaningful regions and overlying them to enhance the grounding ability of GPT-4V. However, for the complicated questions that usually need multi-step information extracting and reasoning, only partitioning the whole image is not promising to improve the reasoning.

In this paper, we propose a simple but powerful multimodal prompting method called UnAC (**Un**derstanding, **A**bstracting and **C**hecking) to improve the abilities of complicated multi-modal

reasoning for LMMs. UnAC consists of a three-step prompting mechanism. In the first step, we present a novel adaptive visual prompting scheme, the second step is abstracting the image into sentences and the final step includes a gradual self-checking prompting. Firstly, to reduce misunderstanding or missing details, the visual prompts are designed as adaptive markers on the image to make LMMs able to focus on specific regions. By looking at the image part by part, LMMs can find more details and have a better understanding of the image overall. Secondly, to solve a problem that needs complex reasoning, we need to correctly abstract the information from the image based on the question. Inspired by the fact that based on the relationship between image and question, humans often extract important information form the image locally and globally. We propose to find the most related parts of the question and abstract the image into language based on the built visual prompts. Then, for a complicated question, the LMMs are easily to make mistakes in some steps, and asking LMMs to check the overall reasoning process is ineffective. However, with the visual context introducing, the checking of a single step is possible. We introduce a gradual self-checking scheme to check each decompsed question individually to improve the accuracy of the answer.

We evaluate UnAC on three datasets of evaluating the ability of complicated problem-solving in the visual context, namely MathVista Lu et al. (2023), MM-Vet Yu et al. (2023) and MMMU Yue et al. (2023). To show the generalization of our method, we conduct experiments on two kinds of LMMs: (a) the powerful and large-scale multimodal models including GPT-4V and Gemini-1.5-flash; (b) relatively light-weighted models including LLaVA-v1.6-7B/13B. We achieve improvements on all models and all datasets which indicates our method is model-agnostic. Notably, our method improves 6.4% on MathVista with Gemini-1.5-flash. To summarize, our main contributions are:

- We propose a simple but powerful multimodal prompting scheme called UnAC (**Un**derstanding, **A**bstracting and **C**hecking) to improve the abilities of complicated multimodal reasoning for LMMs.

- We introduce an adaptive visual prompt to improve the image understanding and reduce the missing details. Combined with the language prompting of the image abstraction and the gradual checking scheme, all the modules lead LMMs to better reasoning.

- Extensive experiments on three datasets which are MathVista, MM-Vet, and MMMU show the effectiveness of UnAC in evoking complicated reasoning in the visual context of LMMs.

## 2 RELATED WORK

**Prompting in LLMs.** We have observed significant advancements in large language models (LLMs) Zhang et al. (2022; 2023); Touvron et al. (2023); Team et al. (2023); Brown et al. (2020). Although the size of LLMs has increased substantially, evoking their reasoning capabilities is still necessary with the use of more complicated designed queries, or prompting. Recently, various works have explored prompt engineering to enhance LLM capabilities. In-context learning has become a mainstream approach to instruct LLMs by providing specific examples Brown et al. (2020); Dong et al. (2022). Building on this, techniques such as chain-of-thought and tree-of-thought Wei et al. (2022); Yao et al. (2024) have been introduced to improve performance in arithmetic, commonsense, and symbolic reasoning tasks. Most recently, Zheng *et al.* Zheng et al. (2023) proposed the Step-Back Prompting method which enhances the ability to retrieve information via abstracting the question. Miao *et al.* Miao et al. (2023) introduced a general-purpose zero-shot verification schema for recognizing errors made in the reasoning process of math problems. However, their methods highly rely on that the language is easy to be decomposed. It is hard to be generalized to the question in the visual context where images are hard to decompose.

**Prompting in LMMs** Before the growth of large multimodal models (LMMs), visual prompting has been explored for various vision and multimodal tasks Wang et al. (2023); Zou et al. (2024); Kirillov et al. (2023); Chen et al. (2022); Shtedritski et al. (2023). These approaches can be categorized into two main types. The first type encodes visual prompts, such as points, boxes, and strokes, into latent features, which are then used to prompt the vision models Zou et al. (2024); Kirillov et al. (2023). The second type overlays visual marks directly onto the input images. These marks can be a red circle Shtedritski et al. (2023), a highlighted region Yang et al. (2023a), or multiple circles with arrows Shtedritski et al. (2023). While these studies show the potential of pixel-level visual prompting, they are typically limited to visually referencing one or a few objects. So far, prompting LMMs has

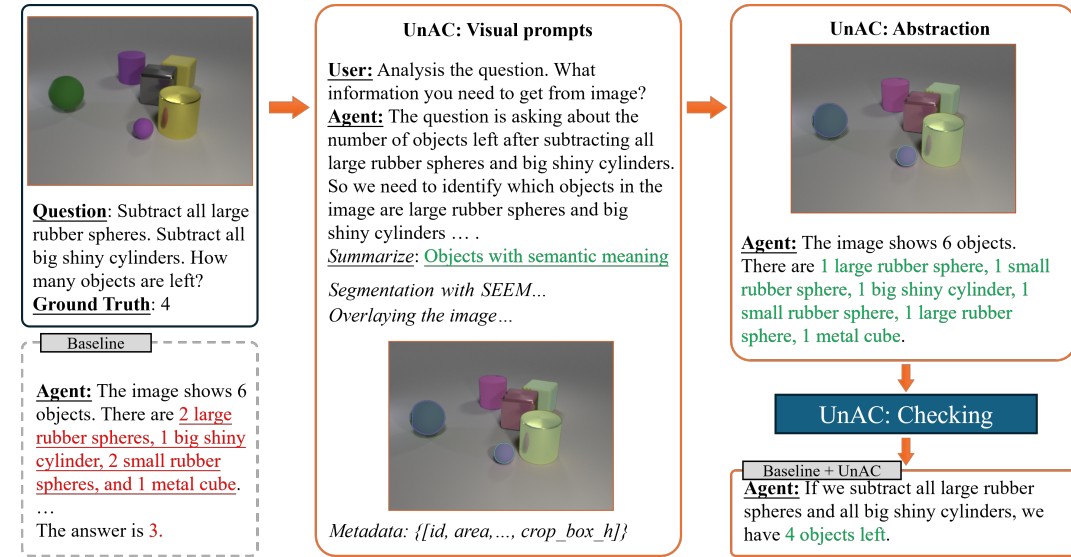

Figure 1: Example of using UnAC. In the original answer from the baseline method, the LMM incorrectly understands and describe the image which leads to the wrong answer. In UnAC which follows in the orange arrows, we first ask the LMM to analyze the question and answer what we need from the image. Then we can summarize the reply as the Objects with semantic meaning. Then employing SEEM to segment and overlay the image as visual prompts. Then abstracting the information of the image where with the markers, the LMM can correctly describe the image and abstract the right contexts. Finally, after the checking stage, we can get the right answer.

been rarely explored in academia, partly because most of the recently open-sourced models have limited capacity and are therefore unable to support such advanced capabilities. Recently, GPT-4V was released, accompanied by a comprehensive qualitative study Yang et al. (2023b). The authors in Yang et al. (2023b) employed a similar prompting strategy as RedCircle Shtedritski et al. (2023) to prompt GPT-4V. Most recently, Yang *et al.* Yang et al. (2023a) proposed to partition the image into a set of semantically meaningful regions and overlay them to enhance the grounding ability of GPT-4V. CCoT Mitra et al. (2023) is designed as a zero-shot Chain-of-Thought prompting method to extract compositional knowledge from an LMM with utilizing scene graphs. However, both of these works can not solve the problem based on the abstract images such as geometry problem solving and math word problems.

## 3 UNAC: UNDERSTANDING, ABSTRACTING, AND CHECKING

Consider a general fact when humans face a challenging problem in the visual context. To solve the problem, we need first to understand the image and the question correctly overall. Then based on the question, we will look at the image more carefully, find and abstract the useful information that can be used to solve the problem. Finally, based on the understanding and the abstraction, we infer the final answer to this challenging problem. Moreover, for a complicated question, we usually need a second look at the reasoning process and check it with the image to avoid some simple mistakes. Inspired by this common sense, we propose UnAC which means understanding, abstracting, and checking for synergizing the complicated reasoning in the visual context of large multimodal models.

### 3.1 ADAPTIVE VISUAL PROMPTS.

Precisely capturing the details in the image is not straightforward for LMMs. It is hard to correct the misunderstanding of the image by itself because decomposing the image is not easy. Since LMMs are developed based on the LLMs, their abilities of language reasoning are much better than visual reasoning. It means that LMMs can perform better on analyzing the problem than analyzing the image. Therefore, we propose to build effective and adaptive multimodal prompts based on the

analysis of the question. Asking the model to analyze the question and find what information we need to get from the image. We conclude the response into two kinds: Objects with semantic meaning and symbols with literal meaning. For objects with semantic meaning, we employ segmentation models to automatically segment the image. For symbols with literal meaning, we use optical character recognition (OCR) methods to detect the texts. Based on the metadata, we first denoising regions based on the stability score output by the segmentation/OCR methods.

In the Figure. 1, we show a successful case. For this question of subtracting the items, it requires LMMs to correctly recognizing each item in the picture which is related to objects with semantic meanings. Therefore, the visual prompts are designed as the segmentation of the image to help the LMM to better understand the image.

## 3.2 IMAGE ABSTRACTION

The visual prompts can make a better understanding of the image since the markers can catch more attentions on some local information. Partitioning the image makes it decomposable when LMMs understand the image. However, only visual prompts have limited improvements for solving complicated problems. Except for understanding the image, LMMs need to correctly abstract the image to filter the useless information to solve the problem. Without prompts of abstraction, the reasoning might be misdirected due to the markers in the image. Therefore, to fully utilize the visual prompts and get better reasoning, we need to abstract the information which is the most related to the question. Firstly, we ask LMMs to describe the picture to abstract the global information. Then based on the analysis of the question and the prompts, we ask LMMs to find the most related regions to get more details based on the markers in the image.

## 3.3 GRADUAL CHECKING

Moreover, for some complicated questions, we usually need a second look at the image with the reasoning progressing. As discussed in Ling et al. (2024), checking the whole reasoning process is usually ineffective for LLMs and our experiments show similar results in LMMs. However, to correct the mistake made in one step is more effective. To check individual steps of the reasoning process, the first thing we should note is that the correctness of each step is highly dependent on its context. For a question in words, the context includes the question and previous steps only. So the checking is largely dependent on the accuracy of the previous steps which is highly unstable. In the visual question answering, the information from the image becomes extra contexts which are important references for self-checking. It can be more reliable when LMMs have a good understanding of the image.

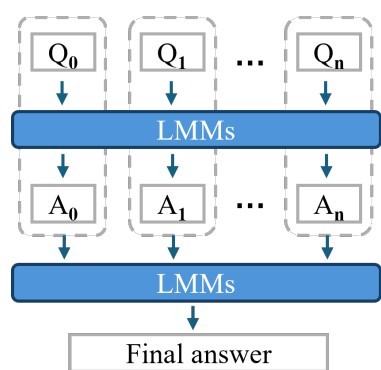

Figure 2: The workflow of the gradually checking.

Then, we design a gradual checking prompting for better reasoning. As illustrated in the Figure. 2, firstly, we let LMMs decompose the question into multi sub-questions $[Q_0, Q_1, \ldots, Q_n]$ and give the answer of each sub-questions. The answers are denoted as $[A_0, A_1, \ldots, A_n]$. In the checking stage, we check gradually. When checking $Q_i$ and $A_i$, we refer the context of the previous questions and checked answers $[Q_0, Q_1, \ldots, Q_i]$ and $[A'_0, A'_1, \ldots, A'_i]$. In the last step of checking, LMMs will infer the final answer based on all questions and answers.

## 4 EXPERIMENTS

Here we define the tasks and models we experiment with. We describe our evaluation metric and the baselines we consider. Moreover, we conduct sufficient experiments and ablations to show the effectiveness and for a better understanding of the behavior of our method.

Table 1: Accuracy scores on the *testmini* subset of MathVista Lu et al. (2023). ALL: overall accuracy. Task types: FQA: figure question answering, GPS: geometry problem solving, MWP: math word problem, TQA: textbook question answering, VQA: visual question answering. Mathematical reasoning types: ALG: algebraic reasoning, ARI: arithmetic reasoning, GEO: geometry reasoning, LOG: logical reasoning, NUM: numeric commonsense, SCI: scientific reasoning, STA: statistical reasoning.

| Method | ALL | FQA | GPS | MWP | TQA | VQA | ALG | ARI | GEO | LOG | NUM | SCI | STA |
|---|---|---|---|---|---|---|---|---|---|---|---|---|---|
| | | *Human performance* | | | | | | | | | | | |
| Human Performance | 60.3 | 59.7 | 48.4 | 73.0 | 63.2 | 55.9 | 50.9 | 59.2 | 51.4 | 40.7 | 53.8 | 64.9 | 63.9 |
| | | *Heuristics baselines* | | | | | | | | | | | |
| Random chance | 17.9 | 18.2 | 21.6 | 3.8 | 19.6 | 26.3 | 21.7 | 14.7 | 20.1 | 13.5 | 8.3 | 17.2 | 16.3 |
| Frequent guess | 26.3 | 22.7 | 34.1 | 20.4 | 31.0 | 24.6 | 33.1 | 18.7 | 31.4 | 24.3 | 19.4 | 32.0 | 20.9 |
| | | *Large Multimodal Models (LMMs)* | | | | | | | | | | | |
| LLaVA-v1.6-7B | 35.9 | 43.1 | 21.2 | 27.4 | 49.3 | 36.3 | 27.4 | 31.1 | 23.7 | 18.9 | 25.0 | 50.0 | 44.1 |
| LLaVA-v1.6-7B + UnAC | **38.5**(+2.6) | 37.9 | 28.4 | 32.3 | 53.8 | 38.5 | 31.6 | 34.8 | 23.7 | 10.8 | 25.7 | 50.0 | 44.1 |
| LLaVA-v1.6-13B | 35.8 | 45.3 | 21.6 | 29.5 | 43.0 | 37.9 | 24.9 | 33.9 | 23.8 | 13.5 | 27.7 | 49.1 | 48.1 |
| LLaVA-v1.6-13B + UnAC | **37.8**(+2.0) | 37.5 | 31.7 | 30.7 | 53.8 | 38.5 | 33.4 | 34.6 | 32.2 | 10.8 | 25.7 | 53.3 | 44.9 |
| Gemini-1.0-pro-vision | 41.0 | 36.4 | 36.5 | 43.0 | 57.5 | 36.3 | 39.8 | 37.6 | 38.0 | 10.8 | 29.8 | 52.4 | 45.5 |
| Gemini-1.0-pro-vision + UnAC | **47.4**(+6.4) | 49.4 | 39.5 | 45.2 | 62.7 | 42.5 | 43.8 | 44.2 | 40.1 | 29.7 | 36.1 | 54.9 | 57.1 |
| GPT-4V | 50.7 | 43.6 | 50.5 | **57.5** | 65.2 | 38.4 | 53.0 | 49.0 | 51.0 | 21.6 | 20.1 | 63.1 | 55.8 |
| GPT-4V + UnAC | **55.6**(+4.9) | 45.3 | 59.1 | 53.2 | 67.7 | **46.9** | 60.9 | **50.1** | 58.5 | 18.9 | 35.4 | 60.7 | 57.8 |
| Gemini-1.5-flash | 53.2 | 51.6 | 56.8 | 52.1 | 67.5 | 40.1 | 59.4 | 44.0 | 52.7 | 25.3 | 36.4 | 60.8 | 57.5 |
| Gemini-1.5-flash + UnAC | **56.6**(+3.4) | **57.3** | **59.3** | 54.1 | **71.6** | 41.4 | **62.8** | 46.6 | **59.5** | **30.9** | **37.7** | **65.3** | **65.8** |

## 4.1 SETUP

**Tasks and datasets.** We experiment with the following two tasks that need complicated reasoning: (a) Mathematical reasoning in the visual context, and (b) Complicated VQA.

- **Mathematical reasoning:** We evaluate MathVista Lu et al. (2023) for this task. MathVista is a consolidated mathematical reasoning benchmark within visual contexts. It contains various kinds of sub-tasks to evaluate the model's visual understanding of mathematical problems solving in different perspectives of reasoning skills.

- **Complicated VQA:** For this task, we evaluate two datasets called: MM-Vet Yu et al. (2023) and MMMU Yue et al. (2023) respectively. MM-Vet Yu et al. (2023) is designed to evaluate large multimodal models on complex multimodal tasks that highlight six core vision-language (VL) capabilities: Recognition, Knowledge, Optical Character Recognition (OCR), Spatial Awareness, Language Generation, and Math. MMMU focuses on advanced perception and reasoning with domain-specific knowledge, challenging models to perform tasks like those faced by experts.

**Models.** To show the generalization of UnAC, we use the following state-of-the-art LLMs: superior models including Gemini-1.0-flash, GPT4-V and Gemini-1.5-flash, relatively small LMMs including LLaVA-v1.6-7B/13B. For the closed-source LMMs, we utilize the official API to make the evaluation. We use 'gpt4-turbo' and 'gemini-1.5-flash' for GPT4-V and Gemini respectively. For the open-source models, we evaluate LLaVA-v1.6-7B/13B in a single RTX 6000. We set the temperature to 0.0 for all LMMs. We use SEEM Zou et al. (2024) for segmentation and easyOCR for building the visual prompts. Studies on these models can be found in the Appendix.

**Evaluation.** In all datasets, they have a unique answer to each question which can be a number, a word, a phrase, or one of the choices. The accuracy (ACC) is the only metric we employed in this paper. Since the LMMs may often generate long-form answers which are hard to capture. Following Lu *et al.* Lu et al. (2023) and Yu *et al.* Yu et al. (2023), we instead conduct an evaluation using the GPT-4 model where we few-shot prompt the model to identify equivalence between target answers and the model predictions.

## 4.2 RESULTS.

**Mathematical reasoning in the visual context.** In Table 1, we show the results on the MethVista Lu et al. (2023). Our method makes improvements on all models. We make an improvement of $4.9\%$ on GPT-4V with our method. For Gemini, we make the largest increase of $6.4\%$ on Gemini-1.0-pro-vision and $3.4\%$ on Gemini-1.5-flash. For LLaVA-v1.6-7B/13B, we achieve the improvements of $2.6\%$ and $2.0\%$ respectively. View in sub-tasks, our method significantly improves the most challenging sub-tasks (*i.e.* Geometry problem solving (GPS) and Math word problem (MWP)). We have a $8.6\%$ improvement compared to the performance of the baseline of GPT-4V on GPS. And $4.1\%$ improvements with Gemini-1.5-flash on TQA. Moreover, UnAC achieves an impressive $13.0\%$ increase on FQA with Gemini-1.0. The improvements on GPS and MWP, which need complex and multi-step reasoning, benefit from the successful abstracting of the information from the image and an effective checking scheme. The improvements on VQA and FQA which need relatively straightforward reasoning show that the adaptive visual prompting scheme helps LMMs understanding the image. The consistent increase of all models shows UnAC is a model-agnostic prompting method.

Moreover, the improvements on GPT4-V and Gemini are much larger than that of LLavA-v1.6-7B/13B. That is because even with the visual prompts, the understanding of the image still highly relies on the capability of the LMMs. Moreover, the one-step self-checking also needs a strong language reasoning ability. If the LMMs cannot decompose the question in a sufficient way, it may cause the misdirection to the wrong answer. We have experimented with this in Sec 4.3.1.

Table 2: Accuracy scores on the MM-Vet and the validation set of MMMU.

| Method | MM-Vet | MMMU |
|---|---|---|
| LLaVA-v1.6-7B | 47.5 | 36.9 |
| LLaVA-v1.6-7B + Ours | **48.5**(+1.0) | **37.4**(+0.5) |
| Gemini-1.5-flash | 62.2 | 56.1 |
| Gemini-1.5-flash + Ours | **64.9**(+2.7) | **60.9**(+4.8) |
| GPT4-V | 67.2 | 57.2 |
| GPT4-V + Ours | **69.3**(+2.1) | **59.7**(+2.5) |

**Complicated VQA.** In Table 2, we show the results on the MM-Vet Yu et al. (2023) and MMMU Yue et al. (2023). In these two datasets, the questions are more generalized with a relatively simple reasoning process. Our method still makes improvements on all models. We make an improvement of $2.5\%$ on GPT-4V with our method and make the largest increase of $4.8\%$ on Gemini-1.5-flash on MMMU. For LLaVA-v1.6-7B/13B, we achieve the improvements of $1.0\%$ on MM-Vet.

The gap between the increase on Gemini/GPT4-V and the increase of LLaVA-v1.6-7B is larger compared to that on MathVista. In these two datasets, they require more comprehensive vision-language capabilities and abundant knowledge reserve on various topics. Therefore, in those two datasets, understanding can be more important than abstracting and reasoning.

## 4.3 ANALYSIS

We now perform some ablations to justify some of the key design choices made by UnAC and provide insights on its behavior. Limited by budget and time, most experiments in this section are performed with Gemini-1.5-flash and LLaVA-v1.6-7B. More experiments on GPT4-V can be found in the Appendix.

### 4.3.1 HOW THE ABSTRACTION AND SELF-CHECKING AFFECT THE FINAL ANSWER?

As we discussed in Sec 4.2, the improvements made by UnAC are influenced by the original capability of the baseline LMMs. Although it makes sense, we want to find out how it influences our method. We conduct experiments on changing the models which is used in abstracting, checking, and final reasoning mainly with LLaVA-v1.6-7B and Gemini-1.5-flash. As shown in Table 3, we replace the

Table 3: The overall accuracy on the *textmini* dataset of MethVista Lu et al. (2023). **L** means LLaVA-v1.6-7B and **G** means Gemini-1.5-flash.

| Abstracting | Checking | Conclusion | ACC |
|:-----------:|:--------:|:----------:|:----:|
| **L** | **L** | **L** | 46.5 |
| **G** | **L** | **L** | 48.4 |
| **L** | **G** | **L** | 48.4 |
| **G** | **G** | **L** | 51.6 |
| **L** | **L** | **G** | 51.6 |
| **G** | **G** | **G** | **56.6** |

LLaVA-v1.6-7B with Gemini-1.5-flash on different roles in our prompting process. Comparing the first Four rows, the final conclusion performs much better when replacing LLaVA-v1.6-7B with Gemini-1.5-flash for performing abstracting, and checking respectively. The best performance is contributed by using Gemini-1.5-flash to make both abstracting and checking among these three ablations. It indicates that better abstracting and checking are helpful for increasing the overall performance. However, comparing the four rows and the bottom row, although Gemini-1.5-flash may provide the accurate answer to the question in the checking stage, the LLaVA-v1.6-7B still infers bad reasoning in the last step. Moreover, comparing the fourth row and fifth row, we can find that even LLaVA-v1.6-7B provides the bad prompts, Gemini-1.5-flash still has the ability of self-correction in conclusion. Although improving the abstracting and checking can lead to better performance, the reasoning abilities of LMMs are still the bottleneck of how well UnAC can perform in solving complicated questions.

### 4.3.2 CORRECTED ERROR ANALYSIS.

Comparing the original predictions of UnAC to the baseline Gemini-1.5-flash model on MathVista and MM-Vet: we find that our methods correct $25.4\%$ errors from the baseline while introducing $5.5\%$ errors on the task of Mathematical reasoning in the visual context. For complicated VQA *i.e.* MM-Vet, UnAC corrects $20.1\%$ errors from the baseline while introducing $6.2\%$ errors. To further understand how UnAC corrects the errors, we annotate all the wrong predictions corrected by our method of baseline methods in the test set, and categorize them into $4$ classes (see Appendix for examples in each class):

- **Misunderstanding**: The error is after introducing the prompts, the LMMs misunderstand the image which is correct in the baseline method.

- **Context loss**: After introducing our method, it causes the missing of some information from the image which does not happen in the baseline answers.

- **Reasoning Error**: The retrieved context is relevant, but the model still fails to reason through the context to arrive at the right answer.

- **Factual Error**: There is at least one factual error when the model recites its own factual knowledge.

**MathVista:** As shown in Figure 3 (left), the major part of the corrected errors are caused by the misunderstanding of about $35\%$. Combined with the corrected $23\%$ context loss, there are totally of about $58\%$ errors rectified by the adaptive visual prompts which help the LMMs better understand the image and capture more details. Otherwise, in some cases, LMMs could successfully understand the image and capture useful information but they perform incorrect reasoning processes or fail to retrieve the needed factual knowledge. Our proposed self-checking prompting scheme can correct those kinds of errors with careful step-by-step checking which has contributed about $42\%$ in the corrected errors.

**MM-Vet:** For corrected errors on MM-Vet which are shown in Figure 3 (right), the $49\%$ percent of errors caused by the misunderstanding are improved by UnAC and $18\%$ errors are caused by

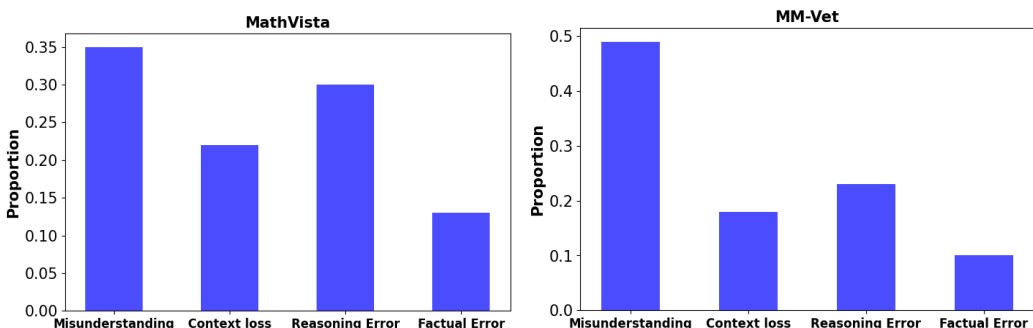

Figure 3: Corrected error analysis of UnAC on MathVista (left) and MM-Vet (right): four classes of errors are corrected by the UnAC

Table 4: Accuracy scores using Gemini-1.5-flash on the *testmini* subset of MathVista Lu et al. (2023). ALL: overall accuracy. Task types: FQA: figure question answering, GPS: geometry problem solving, MWP: math word problem, TQA: textbook question answering, VQA: visual question answering.

| Method | ALL | FQA | GPS | MWP | TQA | VQA |
|---|---|---|---|---|---|---|
| Baseline | 53.2 | 51.6 | 56.8 | 52.1 | 67.5 | 40.1 |
| Segmentation only | 54.2 | 53.2 | 57.8 | 53. | 68.4 | 39.6 |
| OCR only | 53.3 | 51.6 | 57.3 | 51.1 | 68.7 | 40.1 |
| Segmentation + OCR | 54.4 | 53.2 | 57.8 | 53. | 68.4 | 39.6 |
| Adaptive visual prompts | **56.6** | **57.3** | **59.3** | **54.1** | **71.6** | **41.4** |

losing some contexts in the image. Moreover, there are 23% reasoning errors are corrected by the self-checking and 10% are caused by factual loss. In MM-Vet, the questions are relatively more dependent on the understanding of the image. They will ask more about the details in the image which is more important than the reasoning part. Therefore, the visual prompts contribute more effect on correcting the errors.

**Analysis:** Compared to the first two classes and the last two classes, the number of errors removed by correctly understanding the image and capturing more useful contexts is more than that removed by the accurate reasoning process. It indicates that the reasoning step is still a bottleneck of how well UnAC can perform for tasks such as MathVista which requires more complex reasoning.

### 4.3.3 SENSITIVITY TO THE TEMPERATURE

To test the sensitivity to the temperature which has been set to $0.0$ in the paper, we conduct experiments on MathVista Lu et al. (2023) using LLaVA-v1.6-7B and Gemini-1.5-flash as the baseline models. As shown in the Figure. 4, we adjust the temperature form to different values in the section of $[0, 1]$. We can observe that Gemini-1.0-pro-vision is more stable to the changing of the temperatures compared to LLaVA-v1.6-7B. Moreover, for both models, UnAC can make the reasoning more stable under different values of temperatures.

### 4.3.4 WHY DO VISUAL PROMPTS NEED TO BE ADAPTIVE?

In this ablation, we want to show the effect of making the visual prompts adaptive. As shown in Table 4, we conduct experiments on applying different types of visual prompts. Comparing the first two lines, the improvements when employing the segmentation or OCR only are very limited. Although partitions can help the LMMs to focus on a certain part of the image, they also increase the risk of focusing on the wrong regions on the image. Since the whole picture has been overlayed everywhere, it may confuse the attention of LLMs. Adding boxes on the image to let LMMs focus on certain parts, it also increase the risk of incorrect regions which are useless for the question answering. Moreover, for some tasks, markers of segmentation or boxes from OCR is not helpful such as solving a geometry problem or understanding a function plot. Both prompts can not provide much useful information.

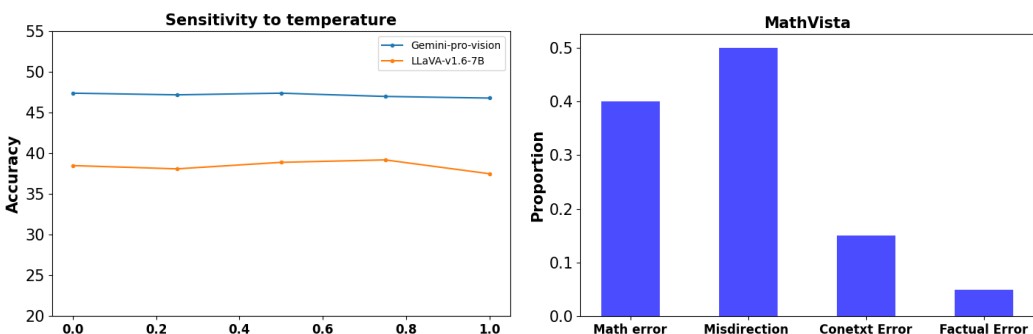

Figure 4: **Left**: The results on MathVista with Gemini-1.0-pro-vision and LLaVA-v1.6-7B under different values of temperatures. **Right**: The error analysis on MethVista with Gemini-1.5-flash using global checking.

Table 5: Accuracy scores using Gemini-1.0-pro-vision on the *testmini* subset of MathVista Lu et al. (2023) under different checking methods. ALL: overall accuracy. Task types: FQA: figure question answering, GPS: geometry problem solving, MWP: math word problem, TQA: textbook question answering, VQA: visual question answering.

| Method | ALL | FQA | GPS | MWP | TQA | VQA |
|---|---|---|---|---|---|---|
| w/o Checking | 45.4 | 47.9 | 36.5 | 44.7 | 57.5 | 41.6 |
| Global Checking | 45.5 | 45.3 | 31.7 | 42.7 | 63.2 | 47.5 |
| One-step Checking | **47.4** | 49.4 | 38.5 | 46.2 | 62.7 | 42.5 |

### 4.3.5 GLOBAL CHECKING OR GRADUAL CHECKING

To prove that LMMs can not perform the global checking in an effective way like LLMs Ling et al. (2024), we conduct experiments on comparing the performance of global checking prompting with the proposed one-step checking method. As shown in Table 5, compared to UnAC without checking, the performance of using the global checking shows very limited improvement overall. Although it increase the accuracy of the textbook question answering and visual question answering tasks, it makes the qualities on tasks of math word problem and geometry problem solving worse. On MWP and GPS tasks, questions are usually more complex reasoning compared to problems in other tasks.

We also conduct experiments on analyzing the errors made by the global checking. As shown in Figure. 4 (right), we define another set of errors which are related to the reasoning process only. The classes of errors are (a) Math error: The additional mathematical errors like computation and mathematical inference; (b) Misdirection: Leading to focusing the wrong regions of the images. (c) Context error: Incorrectly understanding the images or solutions in the previous steps. Misdirection and Math errors are the most frequent errors occurring which have 50% and 39%. It indicates that the global checking easily makes the reasoning process into the wrong direction due to the limitation of the reasoning ability of LMMs.

## 5 DISCUSSION

Compared to the prompting in the LLMs, building an effective prompting method for LMMs can be much harder. The language is easy to decompose since the input and output of texts can be tokenized. However, for images, the input tokens are not necessarily split by semantic. The understanding of image is hard to be decomposed which needs strong capability for dealing with the fine-grained tasks. That's why prompts with language only are also hard to improve the understanding of the images. The LMMs can not successfully process or decompose the image by the order of language.

Nevertheless, visual prompts are neither necessary nor possible to work in all scenarios. For instance, when facing highly abstract problems like geometry problem solving, the understanding of the image mostly depends on the original capability or the trained dataset of the LMMs since even a simple

shape like a heptagon might be misidentified. How to effectively develop visual prompts for such problems is still a challenging topic and that's one of the future works we will target on.

## 6 CONCLUSION

In this paper, we propose a novel multimodal prompting method, namely UnAC (Understanding, Abstracting, and Checking), to synergize reasoning for complicated problems in visual context of LMMs. UnAC consists of an adaptive visual prompting building, the prompts of image abstraction and a gradual checking scheme. Suffecient experiments show the effectiveness of UnAC on improving the ability of complicated multimodal reasoning.

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
