# OpenReview forum: "Understanding, Abstracting and Checking: Evoking Complicated Multimodal Reasoning in LMMs"
_ICLR.cc/2025/Conference — ICLR 2025 Conference Withdrawn Submission_

### Official Review · Reviewer_Pbi6 · 2024-10-30

**Soundness:** 3
**Presentation:** 3
**Contribution:** 2
**Rating:** 5
**Confidence:** 3

**Summary:**

This paper introduces UnAC, an advanced multimodal prompting technique that significantly improves the reasoning capabilities of large multimodal models (LMMs) through adaptive visual prompting, image abstraction, and gradual self-checking, as evidenced by superior performance on MathVista, MM-Vet, and MMMU benchmarks.

**Strengths:**

* The paper introduces the novel UnAC method, combining adaptive visual prompting, image abstraction, and self-checking, which is a new approach for enhancing LMM reasoning.

* The research is thoroughly validated with extensive experiments on MathVista, MM-Vet, and MMMU benchmarks, showing significant performance improvements.


* The paper is well-organized and clearly explains the methods and results, making the innovative contributions easy to understand.

**Weaknesses:**

1. Although this method includes very detailed ablation experiments, it lacks comparisons with other methods.

2. It would be even better if these methods could be validated on test sets that are not OCR and do not require reasoning.

**Questions:**

1. Why is the gradual checking in Fig. 2 a sequential problem? Are the subproblems related to each other?

2. Why does the ADAPTIVE VISUAL PROMPTS phase primarily categorize reasoning problems as segmentation/OCR? Are there other possible methods?

3. If the OCR results are incorrect, are there any remedial measures?

---

### Official Review · Reviewer_6jbw · 2024-11-01

**Soundness:** 2
**Presentation:** 2
**Contribution:** 1
**Rating:** 3
**Confidence:** 3

**Summary:**

This manuscript proposes a multimodal prompting method for large multimodal models (LMMs) named UnAC. UnAC consists of three modules: adaptive visual prompting building, image abstraction and a gradual checking scheme. The experimental results on three distinct benchmarks demonstrate the effectiveness on multiple LMMs, achieving satisfactory improvements.

**Strengths:**

- The paper is easy-to-follow.
- Strong experimental results.

**Weaknesses:**

- Very weak technical contributions. This paper basically borrows some off-the-shelf model to give additional annnotations and then uses existing prompt engineering methods to guide LMM to do reasoning.
- Lack of details on Gradual Checking in Section 3.3. e.g., what if LMMs found the answers are wrong? How does it correct the wrong answers? Figure. 2 did not clarify how this module works and lacked necessary information. It could be improved with some examples.
- Line 352 indicated that UnAC introduced extra errors on the task of Mathematical reasoning, but there is no analysis of these errors. It would be better to report and analyse some failure cases.
- Line 420 stated that UnAC can make the reasoning more stable under different values of temperatures. But in Figure. 4, there is no comparison between models w and w/o UnAC. This statement is not supported by any experimental results in the manuscript.
- Very rush writing. Line 263, 317 and 356 stated that more experiments and examples are **provided in the Appendix. However, there is no appendix in the PDF.**

**Questions:**

- How is OCR specifically implemented in adaptive visual prompts? Does it just add extra boxes as markers for detected texts on the image (Line 429 to 431)? If so, why not utilize the detected texts with their locations as extra text prompts as in MathVista [1]?
- The adaptive visual prompts module helps the model focus on objects with semantic meaning (Line 164), but it is not helpful for problems such as solving a geometry problem or understanding a function plot (Line 431). But in Table 4, the improvements of adopting adaptive visual prompts on figure question answering (FQA), textbook question answering (TQA) and visual question answering (VQA) are quite different. FQA and TQA mainly focus on solving and understanding geometry/chart/function problems, and the improvements are relatively high (5.7% and 4.1%). In contrast, on VQA , which mainly focuses on understanding objects with semantic meaning, the improvements are limited (1.3%). These experimental results are against the aforementioned statements, what is the reason?

---

### Official Review · Reviewer_S8LF · 2024-11-02

**Soundness:** 2
**Presentation:** 3
**Contribution:** 2
**Rating:** 5
**Confidence:** 4

**Summary:**

This work presents UnAC, a multimodal prompting method for complex multimodal problems in large multimodal models. It includes an adaptive visual prompting method for better image understanding, an image abstracting prompting for effective information extraction, and a gradual self-checking scheme for improved reasoning.

**Strengths:**

The paper introduces UnAC, a new multimodal prompting approach. It consists of an adaptive visual prompting method to enhance image understanding, an image abstracting prompting to extract information, and a gradual self-checking scheme for better reasoning. Extensive experiments on three public benchmarks validate the effectiveness of this method.

**Weaknesses:**

1. This paper presents a method that appears overly simple. However, it is claimed to be absolutely effective. Whether it's pure large language models (LLMs) or vision-language models (VLLMs), more context and thought chains have been proven effective multiple times.

2. Some references can be added to the experiments in this paper, such as CCOT:
    Compositional Chain-of-Thought Prompting for Large Multimodal Models. CVPR, 2024.

3. The author conducts experiments only on three public benchmarks (MathVista, MM-Vet, and MMMU). The question is why only these datasets are chosen and what makes them more challenging compared to other datasets.

**Questions:**

See the weaknesses.

---

### Official Review · Reviewer_rWfd · 2024-11-02

**Soundness:** 3
**Presentation:** 3
**Contribution:** 3
**Rating:** 5
**Confidence:** 4

**Summary:**

This paper proposes an adaptive visual prompting method for Multimodal LLMs to boost the visual reasoning in vision-language tasks. Experiments on three public benchmark verify the efficiency of the proposed method.

**Strengths:**

This paper tackles an interesting problem in multimodal LLMs where how to enhance the reasoning ability of the visual context is challenging. The proposed method can lift the reasoning ability of base MLLMs to another level.

**Weaknesses:**

More thorough experiments on base MLLMs such as GPT4o, Claude 3 and Claude 3.5 would be expected.

**Questions:**

Please see weakness.

---

### Note · Authors · 2024-12-02

I have read and agree with the venue's withdrawal policy on behalf of myself and my co-authors.